# Emergent order in the kagome Ising magnet $Dy_3Mg_2Sb_3O_{14}$

Joseph A.M. Paddison[1,2], Harapan S. Ong[1], James O. Hamp[1], Paromita Mukherjee[1], Xiaojian Bai[2], Matthew G. Tucker[3,4], Nicholas P. Butch[5], Claudio Castelnovo[1], Martin Mourigal[2] & S.E. Dutton[1]

The Ising model—in which degrees of freedom (spins) are binary valued (up/down)—is a cornerstone of statistical physics that shows rich behaviour when spins occupy a highly frustrated lattice such as kagome. Here we show that the layered Ising magnet $Dy_3Mg_2Sb_3O_{14}$ hosts an emergent order predicted theoretically for individual kagome layers of in-plane Ising spins. Neutron-scattering and bulk thermomagnetic measurements reveal a phase transition at $\sim 0.3\,K$ from a disordered spin-ice-like regime to an emergent charge ordered state, in which emergent magnetic charge degrees of freedom exhibit three-dimensional order while spins remain partially disordered. Monte Carlo simulations show that an interplay of inter-layer interactions, spin canting and chemical disorder stabilizes this state. Our results establish $Dy_3Mg_2Sb_3O_{14}$ as a tuneable system to study interacting emergent charges arising from kagome Ising frustration.

[1] Department of Physics, Cavendish Laboratory, University of Cambridge, JJ Thomson Avenue, Cambridge CB3 0HE, UK. [2] School of Physics, Georgia Institute of Technology, Atlanta, Georgia 30332, USA. [3] ISIS Neutron and Muon Source, Rutherford Appleton Laboratory, Harwell Campus, Didcot OX11 0QX, UK. [4] Spallation Neutron Source, Oak Ridge National Laboratory, Oak Ridge, Tennessee 37831, USA. [5] NIST Center for Neutron Research, National Institute of Standards and Technology, Gaithersburg, Maryland 20899, USA. Correspondence and requests for materials should be addressed to J.A.M.P. (email: paddison@gatech.edu) or to S.E.D. (email: sed33@cam.ac.uk).

The kagome lattice—a two-dimensional (2D) arrangement of corner-sharing triangles—is at the forefront of the search for exotic states generated by magnetic frustration. Such states have been observed experimentally for Heisenberg[1–4] and planar[5–7] spins. If Ising spins lie within kagome planes and point either towards or away from the centre of each triangle, the potential for emergent behaviour is shown by considering a spin (magnetic dipole) as two separated + and − magnetic charges: the emergent charge $\mathcal{T}$ of a triangle is defined as the algebraic sum over the three charges it contains (Fig. 1a)[8]. Ferromagnetic nearest-neighbour interactions favour $\mathcal{T} = \pm 1$ states, yielding six degenerate states on each triangle. This macroscopic ground-state degeneracy leads to a zero-point entropy $S_0 \approx \frac{1}{3} \ln \frac{9}{2} R$ per mole of Dy (where $R$ is the molar gas constant), and suppresses spin order[9], in analogy to three-dimensional (3D) spin-ice materials[10,11]. The long-range magnetic dipolar interaction generates an effective Coulomb interaction between emergent charges, driving a transition to an emergent charge ordered (ECO) state that is absent for nearest-neighbour interactions alone[8,12]. In this state, + and − charges alternate, but the remaining threefold degeneracy of spin states for each charge means that spin order is only partial (Fig. 1b). The ECO state has two bulk experimental signatures: non-zero entropy $S_0 \approx 0.11R$ per mole of Dy[12], and the presence of both Bragg and diffuse magnetic scattering in neutron-scattering measurements[13,14]. Experimentally, kagome ECO states have been observed in spin-ice materials under applied magnetic field[15,16] and nano-fabricated systems in the 2D limit[14,17–19]. However, a crucial experimental observation has remained elusive—namely, observation of the spatial arrangement of emergent charges in a bulk kagome material.

In this article, we show that an ECO state exists at low temperature in the recently-reported bulk kagome magnet $Dy_3Mg_2Sb_3O_{14}$ (ref. 20). Our experimental evidence derives from neutron-scattering and thermodynamic measurements, while Monte Carlo (MC) simulations reveal that this ECO state is stabilized by a combination of interactions between kagome layers, spin canting out of kagome layers and chemical disorder.

## Results

**Structural and magnetic characterization.** Structural and magnetic characterization suggests that $Dy_3Mg_2Sb_3O_{14}$ (ref. 20) is an ideal candidate for an ECO state. The material crystallizes in a variant of the pyrochlore structure (space group $R\bar{3}m$[20]) in which kagome planes of magnetic $Dy^{3+}$ alternate with triangular layers of non-magnetic $Mg^{2+}$ (Fig. 1c). X-ray and neutron powder diffraction measurements confirm the absence of a structural phase transition to $\lesssim 0.2$ K (Supplementary Figs 1 and 2 and Supplementary Tables 1 and 2) and reveal a small amount of site disorder in our sample, with 6(2)% of Dy kagome sites occupied by Mg (and 18(6)% of Mg sites occupied by Dy). Curie-Weiss fits to the magnetic susceptibility (Fig. 1d) yield a Curie-Weiss constant $\theta_{CW} = -0.1(2)$ K for fitting range $5 \leq T \leq 50$ K, consistent with ref. 20 (however, the value depends strongly on fitting range). Demagnetization effects may also be significant—increasing $\theta_{CW}$ by 1.4 K in spin-ice materials[21]—but cannot be quantitatively determined for a powder sample. The local Dy environment in $Dy_3Mg_2Sb_3O_{14}$ is similar to the cubic spin ice $Dy_2Ti_2O_7$ (ref. 22) (Supplementary Fig. 3), suggesting that $Dy^{3+}$ spins have an Ising anisotropy axis directed into or out of the kagome triangles with an additional component perpendicular to

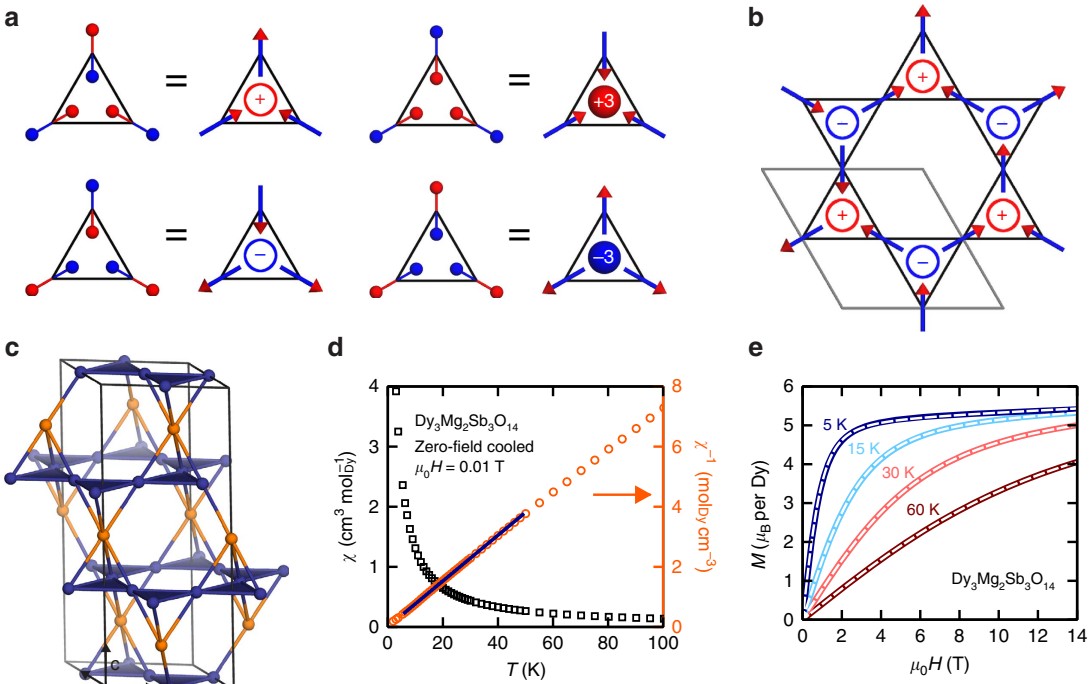

**Figure 1 | Ising spins on the kagome lattice.** (**a**) Relationship between spin vectors (arrows), magnetic dipoles (connected red and blue spheres) and emergent charge $\mathcal{T}$ of a triangle (labelled $\pm$ or $\pm 3$). (**b**) Example of a microstate showing emergent charge order (ECO). (**c**) Partial crystal structure of $Dy_3Mg_2Sb_3O_{14}$, showing kagome $Dy_{1-x}Mg_x$ site (blue spheres) and triangular $Mg_{1-3x}Dy_{3x}$ site (orange spheres), where $x = 0.06(2)$ for the sample of $Dy_3Mg_2Sb_3O_{14}$ studied here. (**d**) Magnetic susceptibility data $\chi(T)$ measured in an applied field $\mu_0H = 0.01$ T after zero-field cooling (left axis; black squares), inverse magnetic susceptibility data $\chi^{-1}$ (right axis; orange circles) and Curie-Weiss fit over the range $5 \leq T \leq 50$ K (blue line). (**e**) Dependence of magnetization $M$ on applied magnetic field $\mu_0H$ at different temperatures (labelled above each curve) and fits to the paramagnetic Ising model. Data are shown as solid coloured lines and fits as white dashed lines (note the nearly perfect agreement: as plotted the fit lines are indistinguishable from the data). In **d,e**, standard errors are derived from fits to the magnetization and are smaller than the symbol size or line width in the plots.

the kagome planes. Experimentally, we confirm Ising anisotropy at low temperatures using isothermal magnetization measurements, which are ideally described by paramagnetic Ising spins with magnetic moment $\mu = 10.17(8)\ \mu_B$ per Dy (Fig. 1e). Moreover, our inelastic neutron-scattering measurements show that the ground-state Kramers doublet is separated from the first excited crystal-field state by at least 270 K (Supplementary Fig. 4), indicating that crystal-field excitations are negligible at the low temperatures ($\leq 50$ K) we consider.

**Low-temperature spin correlations.** The magnetic specific heat $C_m(T)$ shows that spin correlations start to develop below 5 K and culminate in a large anomaly at $T^* = 0.31(1)$ K that we attribute to a magnetic phase transition (Fig. 2a and Supplementary Fig. 5). Below 0.20 K, the spins fall out of equilibrium, as is also reported in spin-ice materials[23]. In zero applied field, the entropy change $\Delta S_m(T)$ from 0.2 K to $T = 10$ K is slightly less than the expected $R\ln 2$ for random Ising spins; however, the full $R\ln 2$ entropy is recovered in a small applied field of 0.5 T. The 0.05(3)$R$ difference between $\Delta S_m(10$ K$)$ in zero field and in a 0.5 T field could be explained either by ECO (with entropy 0.11$R$ in the 2D case[12]), or by the ~6% randomly-oriented orphan Dy spins on the Mg site (with entropy 0.06 $R\ln 2$). Neutron-scattering experiments on a powder sample of $^{162}$Dy$_3$Mg$_2$Sb$_3$O$_{14}$ distinguish these two scenarios by revealing the microscopic processes at play across $T^*$.

Figure 2b shows magnetic neutron-scattering data at 0.5 K (above $T^*$) and at the nominal base temperature of 0.03 K (below $T^*$). At 0.5 K, our data show magnetic diffuse scattering only, with a broad peak centred at $\approx 0.65\ \text{Å}^{-1}$ that is characteristic of ice-rule correlations in structurally related pyrochlore magnets[24]. In contrast, at 0.03 K, strong magnetic diffuse scattering is observed in addition to magnetic Bragg peaks. These peaks develop at $T \leq 0.35$ K; that is, as $T^*$ is crossed. No additional peaks are observed on further cooling and the magnetic scattering does not change between 0.1 and 0.03 K. Between 0.03 and 50 K, the scattering is purely elastic within our maximum experimental resolution of $\approx 17\ \mu$eV (Supplementary Fig. 6), indicating that the spins fluctuate on a timescale longer than $\sim 0.2$ ns. Our 0.03 K data suggest two immediate conclusions. First, the magnetic Bragg peaks are described by the propagation vector $\mathbf{k} = \mathbf{0}$; that is, order preserves the crystallographic unit cell below $T^*$. Second, a large fraction of the magnetic scattering is diffuse; hence, correlated spin disorder persists below $T^*$ and involves the majority of spins. These results cannot be explained by only a small fraction of orphan spins, but are consistent with an ECO state[13,14].

**Average magnetic structure.** We use reverse Monte Carlo (RMC) refinement[25,26] to fit spin microstates to data collected between 0.03 and 4 K. A single RMC microstate can capture both the

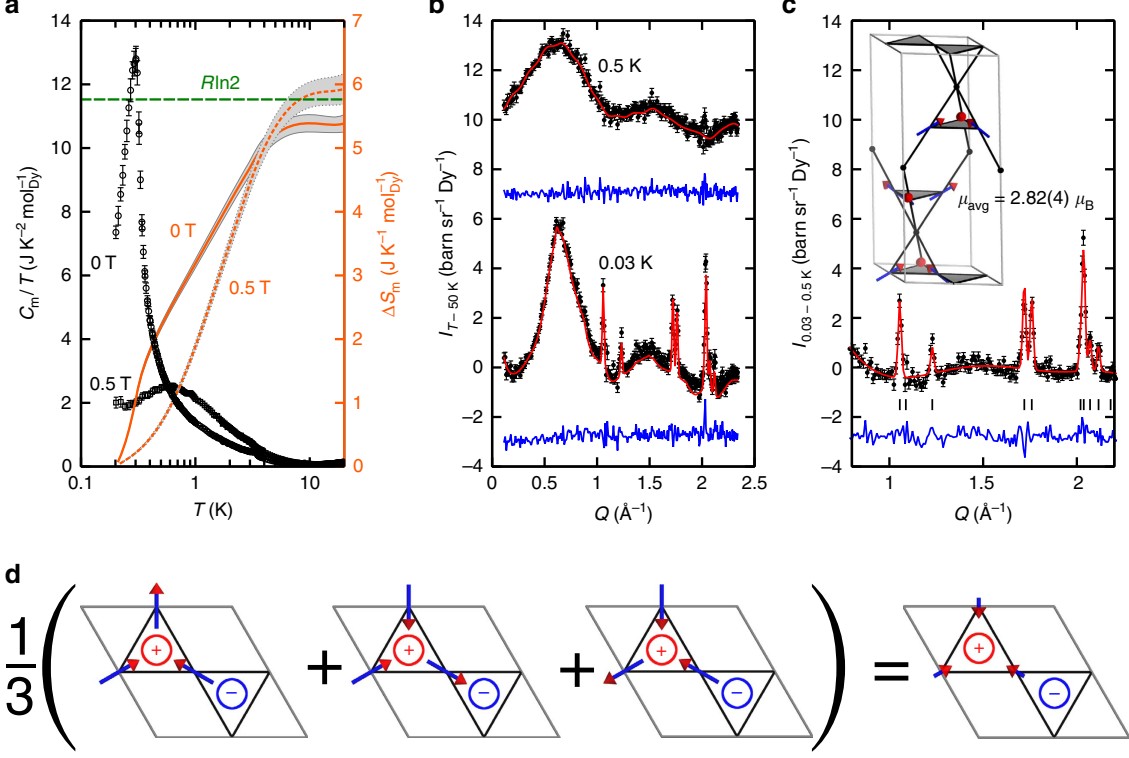

**Figure 2 | Low-temperature magnetism of Dy$_3$Mg$_2$Sb$_3$O$_{14}$.** (**a**) Magnetic heat capacity divided by temperature $C_m/T$ (left axis; black points) and magnetic entropy change $\Delta S_m(T)$ (right axis; orange curves). Zero-field data and data measured in applied field $\mu_0 H = 0.5$ T are shown (fields labelled on each curve). Error bars represent the addition of statistical and systematic uncertainties, where statistical uncertainty is calculated from a least-squares fit of the measured data to a two-timescale relaxation model, and systematic uncertainty is calculated assuming a 5% error on the sample mass. (**b**) Magnetic neutron-scattering data (black circles) at $T = 0.03$ K and 0.5 K obtained by subtracting a high-temperature (50 K) measurement as background, fits from reverse Monte Carlo (RMC) refinements (red lines) and data − fit (blue lines). The 0.5 K curves are vertically shifted by 10 barn sr$^{-1}$ Dy$^{-1}$ for clarity. Error bars on neutron-scattering data indicate one standard error propagated from neutron counts. (**c**) Magnetic Bragg scattering obtained as the difference between 0.03 and 0.5 K data (black circles), fit from Rietveld refinement (red line) and difference (blue line). The inset shows the model of the average magnetic structure obtained from Rietveld refinement. (**d**) The vector average of the three microstates that are equally occupied in a ECO state yields an average all-in/all-out structure with ordered moment $\mu_{avg} = \mu/3$, consistent with experimental observations.

average spin structure responsible for Bragg scattering and the local deviations from the average responsible for diffuse scattering (Fig. 2b and Supplementary Fig. 7). We determine the average spin structure by two methods: first, by averaging refined RMC microstates onto a single unit cell; second, by using a combination of symmetry analysis and Rietveld refinement to model the magnetic Bragg profile (obtained as the difference between 0.03 and 0.5 K data) (Fig. 2c). Details of the Rietveld refinements are given in Supplementary Note 1. Both approaches yield the same all-in/all-out average spin structure (inset to Fig. 2c and Supplementary Fig. 8). The ordered magnetic moment at 0.03 K, $\mu_{avg} = 2.82(4)$ $\mu_B$ per Dy, is much less than the total moment of $\mu \approx 10$ $\mu_B$. These results are consistent with ECO: Fig. 2d shows that averaging over the three possible ECO microstates for a given triangle generates an all-in/all-out average structure, as observed experimentally; moreover, the expected ordered moment for ECO, $\mu/3 \approx 3.3$ $\mu_B$ per Dy[13], is in general agreement with the measured value of 2.82(4) $\mu_B$ per Dy.

**Evidence for emergent charge order.** To look for signatures of ECO in real space, we compare the temperature evolution of $\mu_{avg}$ with the percentage of $\mathcal{T} = \pm 3$ charges (Fig. 3a). The latter quantity, $f_{\pm 3}$, takes a value of 25% for random spins, 100% for an all-in/all-out microstate, and 0% for a microstate that fully obeys the $\mathcal{T} = \pm 1$ ice rule. The value of $f_{\pm 3}$ extracted from RMC refinements decreases with lowering temperature to a minimum value of <5% below 1 K; these values represent upper bounds because RMC refinements were initialized from random microstates. Crucially, below $T^\star$, the $\mathcal{T} = \pm 1$ rule is obeyed while $\mu_{avg}$ is non-zero (Fig. 3a); this coexistence of ice-rule correlations with an all-in/all-out average structure is a defining feature of the ECO state[13,14]. We confirm ECO by calculating the charge-correlation function $\langle \mathcal{T}(0)\mathcal{T}(r_{ab}) \rangle$, the average product of charges separated by radial distance $r_{ab}$ on the honeycomb lattice formed by the triangle midpoints. At 0.5 K, this function decays with increasing $r_{ab}$, indicating that $\mathcal{T} = \pm 1$ charges are disordered (Fig. 3b). At 0.03 K, $\langle \mathcal{T}(0)\mathcal{T}(r_{ab}) \rangle$ shows two key features that indicate an ECO state: a diverging correlation length, and an alternation in sign with a negative peak at the nearest-neighbour distance (Fig. 3c). The magnitude of $\langle \mathcal{T}(0)\mathcal{T}(r_{ab}) \rangle$ found experimentally ($\approx 0.6 = (0.94 \times 3\mu_{avg}/\mu)^2$) is smaller than the value of unity corresponding to an ideal ECO state, which indicates that the alternation of charges contains some errors; we show below this is probably due to the presence of site disorder.

**Explanation of emergent charge order.** Why does $Dy_3Mg_2Sb_3O_{14}$ show fundamentally the same ECO as predicted for a 2D kagome system of in-plane Ising spins? This is far from obvious, because the real material differs from the existing model[8] in three respects: (i) the spins are canted at an angle of $26(2)°$ to the kagome planes, (ii) the planes are layered in 3D and (iii) there is Dy/Mg site disorder (Fig. 2c). This puzzle is elucidated by Monte Carlo simulations for a minimal model containing the nearest-neighbour exchange interaction $J = -3.72$ K determined for structurally-related $Dy_2Ti_2O_7$ (refs 22,27), and the long-range magnetic dipolar interaction $D = 1.28$ K calculated from experimentally determined Dy–Dy distances. In 2D, spin canting interpolates between two limits—an ECO transition followed by lower-temperature spin ordering for in-plane spins[8], and a single spin-ordering transition for spins perpendicular to kagome planes[28]—and hence destabilizes ECO compared with the 2D in-plane limit. In contrast, the stacking of kagome planes stabilizes 3D ECO—uniquely minimizing the effective Coulomb interaction between emergent charges—but leaves the spin-ordering transition temperature essentially unchanged. The

effect of random site disorder is shown in Fig. 3d. Disorder broadens the specific-heat anomalies and suppresses the ECO transition temperature. In spite of this, we find that a distinct ECO phase persists for 6% Mg on the Dy site; that is, the estimated level of disorder present in our sample of $Dy_3Mg_2Sb_3O_{14}$. Moreover, simulated magnetic specific-heat (Fig. 3d) and powder neutron-scattering (Supplementary Fig. 9) curves with ~4 to 6% Mg on the Dy site show remarkably good agreement with experimental data, especially given that $J$ is not optimized for $Dy_3Mg_2Sb_3O_{14}$.

**Implications of emergent charge order.** An ECO microstate can be coarse-grained into a magnetization field with two components: the all-in/all-out average spin structure with non-zero divergence, and the local fluctuations from the average that are captured by (divergence-free) dimer configurations on the dual honeycomb lattice[13]. These two components are independent, which leads to descriptions of the ECO state in terms of spin fragmentation[13,14]. Without site disorder, the fluctuating component yields pinch-point features in single-crystal diffuse-scattering patterns, the signature of a Coulomb phase[13,29]. Figure 3e shows that the introduction of site disorder blurs the pinch points and reduces the magnitude of the ordered moment in the ECO phase. We find good overall agreement between patterns from model simulations with ~4 to 6% Mg on the Dy site and from RMC microstates refined to powder data (Fig. 3e). These results suggest that pinch-point scattering could be observed in single-crystal samples of $Dy_3Mg_2Sb_3O_{14}$ with low levels of disorder. Our simulations also suggest why a transition from ECO to spin ordering is not observed experimentally: single-spin-flip dynamics (arguably more appropriate to real materials) become frozen in the ECO state and non-local (loop) dynamics are required to observe the spin-ordering transition in Monte Carlo simulations.

## Discussion

The ECO state in $Dy_3Mg_2Sb_3O_{14}$ is the first realization of ordering of emergent degrees of freedom in a solid-state kagome material. Phase transitions driven by emergent excitations are rare—related examples being the critical end-point in spin ice[11,30,31] and the recent report of spin fragmentation in pyrochlore $Nd_2Zr_2O_7$ (ref. 32). Moreover, the unusually slow spin dynamics offer the exciting possibility of measuring finite-time (Kibble-Zurek) scaling at the ECO critical point[31]. The ECO state in $Dy_3Mg_2Sb_3O_{14}$ presents an intriguing comparison with other partially ordered magnets. In $Gd_2Ti_2O_7$, symmetry breaking yields two inequivalent Gd sites, only one of which orders[33,34]; in contrast, in the ECO state, all spins possess both ordered and disordered components. In $Ho_3Ga_5O_{12}$, local antiferromagnetic correlations coexist with average antiferromagnetic order[35], whereas in the ECO state, the average order is antiferromagnetic (all-in/all-out) while the local correlations are ferromagnetic (two-in/one-out or *vice versa*). Whether the predicted spin-ordering[8] eventually occurs in $Dy_3Mg_2Sb_3O_{14}$ remains to be seen: spin freezing[36,37] or site disorder may prevent its onset. We expect physical and/or chemical perturbations to control the properties of $Dy_3Mg_2Sb_3O_{14}$; for example, application of magnetic field slightly tilted from the $c$-axis should drive a Kastelyn transition towards spin-ordering[15,16]; modified synthesis conditions may allow the degree of site mixing to be controlled[20]; and application of chemical pressure may alter the spin-canting angle and/or the distance between kagome layers, potentially generating a novel spin-ordering phase instead of ECO for sufficiently large canting[28]. Substitution of $Dy^{3+}$ by other lanthanide ions[20,38–40] may increase the ratio of exchange to

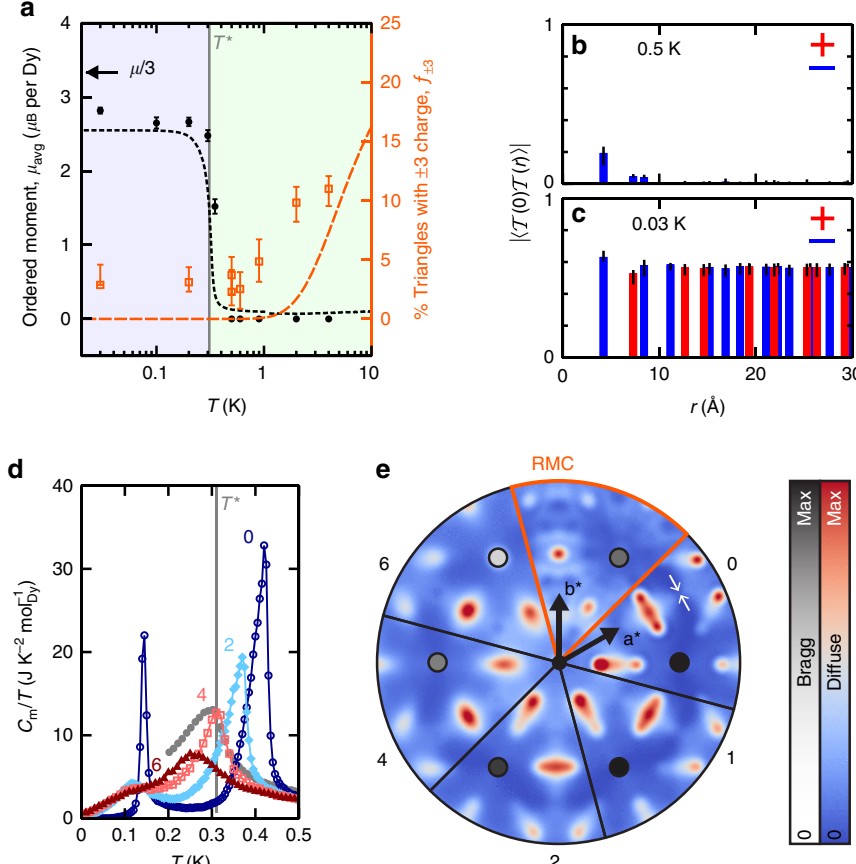

**Figure 3 | Emergent charge order in Dy₃Mg₂Sb₃O₁₄.** (**a**) Temperature evolution of the ordered magnetic moment per Dy, $\mu_{avg}$ (left axis) and the number of triangles for which $\mathcal{T} = \pm 3$, $f_{\pm 3}$ (right axis). Values of $\mu_{avg}$ from Rietveld refinements to experimental data are shown as filled black circles, and values of $\mu_{avg}$ from Monte Carlo (MC) simulations (with 4% Mg on the Dy site) are shown as a black dotted line. Upper bounds on $f_{\pm 3}$ from reverse Monte Carlo (RMC) refinements to experimental data are shown as hollow orange squares, and values from MC simulations as an orange dashed line. The location of $T^*$ is shown by a vertical grey line, and the background is shaded blue below $T^*$ and green above $T^*$. Throughout, error bars for results from Rietveld refinements indicate one standard error from least-squares fitting, and error bars from RMC refinements are derived by assuming 10% uncertainty on the absolute intensity normalization of the magnetic scattering data. (**b**) Charge-correlation function $\langle \mathcal{T}(0)\mathcal{T}(r_{ab})\rangle$ obtained from RMC refinements at 0.5 K, and (**c**) $\langle \mathcal{T}(0)\mathcal{T}(r_{ab})\rangle$ from RMC at 0.03 K. Solid bars show correlation magnitudes, with positive correlations shown in red and negative correlations in blue. (**d**) Magnetic heat capacity from MC simulations (system size $N = 7,776$ spins) for different amounts of random site disorder (the % Mg on the Dy site is labelled above each curve). The uncertainty in the MC results was assessed by computing the standard deviation of statistically-independent simulations; standard errors are smaller than the symbols in the figures. (**e**) Single-crystal neutron scattering calculations in the $(hk0)$ plane from MC simulations at $T = 0.2$ K for different amounts of random site disorder (the % Mg on the Dy site is labelled on each segment of the plot). The single-crystal calculation from RMC refinement to 0.03 K powder data (for 6% Mg on the Dy site) is shown for comparison. Separate colour scales are used for the intensity of the diffuse scattering and the {110} Bragg peaks, and the location of a pinch point is indicated by small white arrows.

dipolar interactions, offering promising routes towards exotic spin-liquid behaviour: dimensionality reduction by effective layer decoupling (when exchange dominates over dipolar interactions), and realization of quantum kagome systems with local spin anisotropies.

## Methods

**Sample preparation.** Powder samples of Dy₃Mg₂Sb₃O₁₄ were prepared from a stoichiometric mixture of dysprosium (III) oxide (99.99%, Alfa Aesar*), magnesium oxide (99.998%, Alfa Aesar*) and antimony (V) oxide (99.998%, Alfa Aesar*). For neutron-scattering experiments a ~5 g sample isotopically enriched with ¹⁶²Dy (94.4(2)% ¹⁶²Dy₂O₃, CK Isotopes*) was prepared. For all samples, starting materials were intimately mixed and pressed into pellets before heating at 1,350 °C for 24 h in air. This heating step was repeated until the amount of impurity phases as determined by X-ray diffraction was no longer reduced on heating. The enriched sample contained impurity phases of MgSb₂O₆ (6.4(5) wt%) and Dy₃SbO₇ (0.97(8) wt%), the latter of which orders antiferromagnetically at $T \approx 3$ K (ref. 41).

*The name of a commercial product or trade name does not imply endorsement or recommendation by the National Institute of Standards and Technology (NIST).

**X-ray diffraction measurements.** Powder X-ray diffraction was carried out using a Panalytical Empyrean* diffractometer with Cu $K\alpha$ radiation ($\lambda = 1.5418$ Å). Measurements were taken between $5 \leq 2\theta \leq 120°$ with $\Delta 2\theta = 0.02°$.

*The name of a commercial product or trade name does not imply endorsement or recommendation by NIST.

**Neutron-scattering measurements.** Powder neutron diffraction measurements were carried out on the General Materials (GEM) diffractometer at the ISIS Neutron and Muon Source, Harwell, UK[42], at $T = 0.50, 0.60, 0.90, 2.0, 4.0, 25$ and 300 K. For $T = 25$ and 300 K measurements, around 4.2 g of isotopically enriched powder was loaded into a $\phi = 6$ mm vanadium can and cooled in a flow cryostat. For measurements at $T \leq 25$ K, the same sample was loaded into a $\phi = 6$ mm vanadium can, which was attached directly to a dilution refrigerator probe and loaded within a flow cryostat. Inelastic neutron-scattering experiments were carried out on the Disk Chopper Spectrometer (DCS) at the NIST Center for Neutron Research, Gaithersburg MD, USA[43], at $T = 0.03, 0.10, 0.20, 0.30, 0.35$, and 0.50 K. Around 1.1 g of isotopically enriched powder was loaded into a $\phi = 4.7$ mm copper can and mounted at the base of a dilution refrigerator. The temperature was measured at the mixing chamber and does not necessarily reflect the sample temperature for 0.1 and 0.03 K, as the spins progressively fall out of equilibrium. On DCS, data were measured with incident wavelengths of 1.8, 5 and 10 Å. The

1.8 Å data were used to look for crystal-field excitations (Supplementary Fig. 4). The 10 Å data were used to look for low-energy quasi-elastic scattering (Supplementary Fig. 6). The 5 Å data were integrated over the energy range $-0.15 \leq E \leq 0.15$ meV to obtain the total scattering (Supplementary Fig. 10). Data reduction was performed using the MANTID and DAVE[44] programs. All data were corrected for detector efficiency using a vanadium standard, normalized to beam current (GEM) or incident beam monitor (DCS), and corrected for absorption by the sample.

**Crystal-structure refinements.** Combined Rietveld analysis of the 300 K X-ray and neutron (GEM) diffraction data was carried out using the FULLPROF suite of programs[45]. The individual patterns were weighted so that the total contribution from X-ray and neutron diffraction was equal; that is, data from each of the five detector banks on GEM was assigned 20% of the weighting of the single X-ray pattern. The neutron scattering cross-section for Dy was fixed to $b_{Dy} = -0.6040$ fm, to reflect the isotopic composition as determined by inductively coupled plasma mass spectrometry. Peak shapes were modelled using a pseudo-Voigt function, convoluted with an Ikeda-Carpenter function or an axial divergence asymmetry function for neutron and X-ray data, respectively. Backgrounds were fitted using a Chebyshev polynomial function. At 25 K, Rietveld analysis of only the neutron diffraction data was carried out. In addition to the impurity phases observed in X-ray diffraction, a small amount ($<1$ wt%) of vanadium (IV) oxide from corrosion of the vanadium sample can was also observed in the neutron-diffraction data. The fit to 300 K data is shown in Supplementary Fig. 1, refined values of structural parameters are given in Supplementary Table 1, and selected bond lengths are given in Supplementary Table 2.

**Magnetic measurements.** Magnetic susceptibility measurements, $\chi(T) = M(T)/H$, were made using a Quantum Design* Magnetic Properties Measurement System (MPMS) with a superconducting interference device (SQUID) magnetometer. Measurements were made after cooling in zero field (ZFC) and in the measuring field (FC) of $\mu_0 H = 0.1$ T over the temperature range $2 \leq T \leq 300$ K. Isothermal magnetization $M(H)$ measurements were made using a Quantum Design* Physical Properties Measurement System (PPMS) at selected temperatures $1.6 \leq T \leq 80$ K between $-14 \leq \mu_0 H \leq 14$ T. A global fit to the $M(H)$ data for $T \geq 5$ K (Fig. 1e) was performed using the powder-averaged form for free Ising spins,

$$M_{Ising}^{powder} = \frac{\mu}{2} \int_{-1}^{1} \cos(\theta) \tanh\left(\frac{\mu H \cos\theta}{k_B T}\right) d(\cos\theta), \quad (1)$$

where $H$ is applied magnetic field, and magnetic moment $\mu$ is the only fitting parameter[21]. The fitted value $\mu = 10.17(8) \mu_B$ per Dy is in close agreement with the expected value of $10.0 \mu_B$ for a Kramers doublet ground state with $g = 4/3$ and $m_J = \pm 15/2$; in particular, the reduced value of the saturated magnetization, $M_{sat} \approx \mu/2$, is as expected for powder-averaged Ising spins[21].

*The name of a commercial product or trade name does not imply endorsement or recommendation by NIST.

**Heat-capacity measurements.** Heat-capacity measurements were carried out on a Quantum Design* Physical Properties Measurement System instrument using dilution fridge ($0.07 \leq T \leq 4$ K) and standard ($1.6 \leq T \leq 250$ K) probes in a range of measuring fields, $0 \leq \mu_0 H \leq 0.5$ T. To ensure sample thermalization at low temperatures, measurements were made on pellets of $Dy_3Mg_2Sb_3O_{14}$ mixed with an equal mass of silver powder, the contribution of which was measured separately and subtracted to obtain $C_p$. The magnetic specific heat $C_m$ was obtained by subtracting modelled lattice $C_l$ and nuclear $C_n$ contributions from $C_p$. We obtained $C_l$ by fitting an empirical Debye model to the $10 < T < 200$ K data, with $\theta_D = 272(13)$ K. To obtain a lower bound on the contact hyperfine and electronic quadrupolar contributions to $C_p$[23,46], we used previous experimental results on dysprosium gallium garnet[47], a related material for which these contributions are known down to $T = 0.037$ K. Correcting for the larger static electronic moment $\approx 4.2 \mu_B$ of dysprosium gallium garnet compared with $\langle \mu \rangle \geq 2.5 \mu_B$ below 0.2 K for $Dy_3Mg_2Sb_3O_{14}$, we obtained the high-temperature tail of the nuclear hyperfine contributions as $C_p = A/T^2$ with $A = 0.0032$ J K mol$_{Dy}^{-1}$ (Supplementary Fig. 5).

*The name of a commercial product or trade name does not imply endorsement or recommendation by NIST.

**Average magnetic structure analysis.** The magnetic Bragg profile was obtained by subtracting data collected at $T < 0.5$ K from the 0.5 K data. Refinements were carried out using the Rietveld method within the FULLPROF suite of programs[45], as described above. For the magnetic-structure refinement shown in Fig. 2c, candidate magnetic structures were determined using symmetry analysis[48] via the SARAH[49] and ISODISTORT[50] programs, as described in Supplementary Note 1. The average magnetic structure is described by the irreducible representation $\Gamma_3$, in Kovalev's notation[51]. The basis vectors of the magnetic structure are given in Supplementary Table 3 and refined values of structural parameters are given in Supplementary Table 4.

**Magnetic total scattering.** To isolate the total magnetic contribution to the neutron-scattering data, data collected at a high temperature $T_{high} \gg \theta_{CW}$ were subtracted from the low-temperature data of interest, where $T_{high} = 25$ K (GEM data) or 50 K (DCS data). For the data obtained below the magnetic ordering temperature of the $Dy_3SbO_7$ impurity phase ($\approx 3$ K (ref. 41)), a refined model of the magnetic Bragg scattering of $Dy_3SbO_7$ was subtracted, as described in Supplementary Note 2 (we note that the orthorhombic crystal structure of $Dy_3SbO_7$ (ref. 52) allowed the impurity Bragg peaks to be readily distinguished from sample peaks). The fit to neutron data of the $Dy_3SbO_7$ magnetic-structure model is shown in Supplementary Fig. 11, the magnetic basis vectors are given in Supplementary Table 5, and refined values of structural parameters are given in Supplementary Table 6. The data were placed on an absolute intensity scale (barn sr$^{-1}$ Dy$^{-1}$) by normalization to the calculated nuclear Bragg profile at $T_{high}$.

**Reverse Monte Carlo refinements.** Refinements to the total (Bragg + diffuse) magnetic scattering were performed using a modified version of the SPINVERT program[53] available from J.A.M.P. In these refinements, a microstate was generated as a periodic supercell containing $N = 7776$ $Dy^{3+}$ spin vectors $S_i = \mu \sigma_i \hat{e}_i$, where $\mu = 10.0 \mu_B$ is the fixed magnetic moment length, the unit vector $\hat{e}_i$ specifies the local Ising axis determined from Rietveld refinement, and the Ising variable $\sigma_i = \pm 1$. A random site-disorder model with 6% non-magnetic Mg on the Dy site was assumed, and $S_i \equiv 0$ for atomic positions occupied by Mg. Ising variables were initially assigned at random, and then refined against experimental data in order to minimize the sum of squared residuals,

$$\chi^2 = W \sum_Q \left[\frac{I_{calc}(Q) - I_{expt}(Q)}{\sigma(Q)}\right]^2, \quad (2)$$

where $I(Q)$ is the magnetic total-scattering intensity at $Q$, subscripts 'calc' and 'expt' denote calculated and experimental intensities, respectively, $\sigma(Q)$ is an experimental uncertainty, and $W$ is an empirical weighting factor. For data collected on GEM, a refined flat-in-$Q$ background term was included in the calculated $I(Q)$. For data collected at $T \leq 0.35$ K, we obtain $I_{calc}(Q) = I_{Bragg}(Q) + I_{diffuse}(Q) - I_{random}(Q)$, where subscripts 'Bragg', 'diffuse' and 'random' indicate magnetic Bragg, magnetic diffuse and high-temperature contributions, respectively. Here, $I_{random}(Q) = \frac{2}{3}C[\mu f(Q)/\mu_B]^2$, where the constant $C = (\gamma_n r_e/2)^2 = 0.07265$ barn and $f(Q)$ is the $Dy^{3+}$ magnetic form factor[54]. The Bragg and diffuse contributions were separated by applying the identity $S_i \equiv \langle S_i \rangle + \Delta S_i$ to each atomic position[55], where the average spin direction $\langle S_i \rangle$ is obtained by vector averaging the supercell onto a single unit cell, and the local spin fluctuation $\Delta S_i \equiv S_i - \langle S_i \rangle$. The Bragg contribution is given by

$$I_{Bragg}(Q) = C\left[\frac{f(Q)}{\mu_B}\right]^2 \frac{2\pi^2 N_c}{NV} \sum_G \frac{|F^\perp(G)|^2}{G^2} R(Q - G), \quad (3)$$

in which $G$ is a reciprocal lattice vector with length $G$, $V$ is the volume of the unit cell, $N_c$ is number of unit cells in the supercell, $R(Q - G)$ is the resolution function determined from Rietveld refinement[56]. The magnetic structure factor $F^\perp(G) = \sum_i \langle S_i \rangle^\perp \exp(iG \cdot r_i)$, where superscript '$\perp$' indicates projection perpendicular to $G$, and the sum runs over all atomic positions in the unit cell. The diffuse contribution is given by

$$I_{diffuse}(Q) = C\left[\frac{f(Q)}{\mu_B}\right]^2 \frac{1}{N} \left\{\frac{2}{3}\sum_i |\Delta S_i|^2 + \sum_{j \neq i}\left[A_{ij}\frac{\sin Qr_{ij}}{Qr_{ij}} + B_{ij}\left(\frac{\sin Qr_{ij}}{(Qr_{ij})^3} - \frac{\cos Qr_{ij}}{(Qr_{ij})^2}\right)\right]\right\}, \quad (4)$$

where sums run over all atomic positions in the supercell, $r_{ij}$ is the radial distance between positions $i$ and $j$, and the correlation coefficients $A_{ij} = \Delta S_i \cdot \Delta S_j - (\Delta S_i \cdot r_{ij})(\Delta S_j \cdot r_{ij})/r_{ij}^2$ and $B_{ij} = 3(\Delta S_i \cdot r_{ij})(\Delta S_j \cdot r_{ij})/r_{ij}^2 - \Delta S_i \cdot \Delta S_j$ (refs 53,57). For data collected at $T \geq 0.5$ K, which show no magnetic Bragg scattering, we obtain $I_{calc}(Q) = I_{diffuse}(Q) - I_{random}(Q)$, with $S_i$ replaces $\Delta S_i$ everywhere. All refinements employed the Metropolis algorithm with single-spin flip dynamics, and were performed for 200 proposed flips per spin, after which no significant reduction in $\chi^2$ was observed. Fits-to-data at $T = 0.03$, 0.20, 0.50, 0.60, 0.90, 2.0 and 4.0 K are shown in Supplementary Fig. 7.

**Monte Carlo simulations.** Simulations were performed for the dipolar spin ice model[27,58], extended to the geometry of interest in this work. The model is defined for Ising spins $S_i = \mu \sigma_i \hat{e}_i$, which are constrained to point along the local easy-axis directions $\hat{e}_i$ and can thus be described by the Ising pseudospin variables, $\sigma_i = \pm 1$. The Hamiltonian comprises an exchange term of strength $J$ between nearest-neighbour spins $\langle i, j \rangle$, and long-range dipolar interactions of characteristic strength $D = (\mu_0/4\pi)\mu^2/r_{nn}^3$ between all pairs of spins, where $\mu \approx 10 \mu_B$ is the magnitude of the $Dy^{3+}$ spin and $r_{nn}$ is the nearest-neighbour distance of the lattice. The Hamiltonian is thus given by

$$\mathcal{H} = -J\sum_{\langle i,j \rangle} \sigma_i \sigma_j (\hat{e}_i \cdot \hat{e}_j) + Dr_{nn}^3 \sum_{i>j} \sigma_i \sigma_j \left(\frac{\hat{e}_i \cdot \hat{e}_j}{r_{ij}^3} - \frac{3(\hat{e}_i \cdot r_{ij})(\hat{e}_j \cdot r_{ij})}{r_{ij}^5}\right), \quad (5)$$

where $\mathbf{r}_{ij}$ is the vector of length $r_{ij}$ connecting spins $i$ and $j$. We use $D = 1.28$ K as calculated from experimentally determined Dy-Dy distances, and $J = -3.72$ K from $Dy_2Ti_2O_7$ (ref. 27), which has a similar Dy environment to $Dy_3Mg_2Sb_3O_{14}$ (ref. 22) (Supplementary Fig. 3). We treat the long-range dipolar interactions using Ewald summation[58,59] with tinfoil boundary conditions at infinity. In simulations including site disorder, non-magnetic ions are simulated by setting the corresponding $\sigma_i$ to zero. Our unit cell comprises three stacked kagome layers, each layer made from four kagome triangles. The whole system comprises $N = 7776$ spins in total, commensurate with the possible $\sqrt{3} \times \sqrt{3}$ spin-ordered state found in 2D (ref. 8). We use both single-spin flip and loop dynamics[58,60], with Metropolis weights. Loop dynamics are necessary to ensure ergodicity at low temperatures and explore possible long-range spin-ordered states. We use the short loop algorithm[58,60]. One Monte Carlo sweep is defined as $N$ single spin-flip attempts, followed by the proposal of loop moves until the cumulative number of proposed spin-flips (in the loops) is at least $N$. We use an annealing protocol, initializing the system at high temperature with $\sim 10^4 N$ single spin-flip attempts, then decrease the temperature incrementally. After each temperature decrement, the system is updated with $\sim 10^3$ Monte Carlo sweeps to ensure equilibration before collecting data every $\sim 10$ Monte Carlo sweeps. Powder-averaged magnetic neutron-scattering patterns calculated from Monte Carlo are shown in Supplementary Fig. 9.

**Data availability.** The underlying research materials can be accessed at the following location: http://dx.doi.org/10.17863/CAM.4902.

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

## Acknowledgements

Work at Cambridge was supported through the Winton Programme for the Physics of Sustainability. The work of J.A.M.P., X.B. and M.M. and facilities at Georgia Tech were supported by the College of Sciences through M.M. start-up funds. J.A.M.P. gratefully acknowledges Churchill College, Cambridge for the provision of a Junior Research Fellowship. H.S.O. acknowledges a Teaching Scholarship (Overseas) from the Ministry of Education, Singapore. J.O.H. is grateful to the Engineering and Physical Sciences Research Council (EPSRC) for funding. C.C. was supported by EPSRC Grant No. EP/G049394/1, and the EPSRC NetworkPlus on 'Emergence and Physics far from Equilibrium'. Experiments at the ISIS Pulsed Neutron and Muon Source were supported by a beamtime allocation from the Science and Technology Facilities Council. This work utilized facilities at the NIST Center for Neutron Research. Monte Carlo simulations were performed using the Darwin Supercomputer of the University of Cambridge High Performance Computing Service (http://www.hpc.cam.ac.uk/) and the ARCHER UK National Supercomputing Service (http://www.archer.ac.uk/, for which access was provided by an ARCHER Instant Access scheme). We thank G.-W. Chern, J. Goff, A. L. Goodwin, G. Lonzarich, G. Moller, D. Prabhakaran, J. R. Stewart and A. Zangwill for valuable discussions, and M. Kwasigroch for preliminary theoretical work.

## Author contributions

H.S.O., P.M. and S.E.D prepared the samples. H.S.O., P.M., X.B., M.M. and S.E.D. performed and analysed the thermo-magnetic measurements. J.A.M.P., P.M., X.B., M.G.T., N.P.B. and S.E.D. performed the neutron-scattering measurements and J.A.M.P., M.M. and S.E.D. analysed the data. J.A.M.P. carried out the RMC refinements. J.O.H. and C.C. carried out the Monte Carlo simulations. C.C. and S.E.D. conceived the project, which was supervised by C.C., M.M. and S.E.D. J.A.M.P. wrote the paper with input from all authors.

## Additional information

**Competing financial interests:** The authors declare no competing financial interests.

**Publisher's note**: 

