## [Peer Review File · Nature Communications]

REVIEWERS' COMMENTS:

Reviewer #1 (Remarks to the Author):

Dear Editor,

I have read carefully the authors response. I think that they address in this new version all my previous comments in a fair and convincing manner. They now put their work in a larger context, showing that the fragmentation mechanism is also at play in bulk Kagome materials. Together with the work presented in Ref 13,14 and 32, this study forms a very interesting set of new results in the field of frustrated magnetism, that should lead to new experimental and theoretical developments. I thus support the publication of this new version.

Reviewer #2 (Remarks to the Author):

Dear editor:

I have had the opportunity to review the revised manuscript by Paddison et al "Emergent Order in the Kagome Ising Magnet $Dy_3Mg_2Sb_3O_{14}$." I am satisfied by the response to the referee reports and I agree that the manuscript should be transferred to Nature Communications. One of the referees point out that there is "no new physics" in this work due to previous work that just came out by Dun et al, but I disagree with this statement. It is unusual to observe emergent charge ordering in a kagome material, and the neutron scattering results presented here are important for showing this explicitly. I recommend publication in Nature Communications.

RESPONSE TO REFEREES:

We are delighted that both referees recommend publication in Nature Communications without any further changes, as shown by their complete reports:

Reviewer 1:

I have read carefully the authors response. I think that they address in this new version all my previous comments in a fair and convincing manner. They now put their work in a larger context, showing that the fragmentation mechanism is also at play in bulk Kagome materials. Together with the work presented in Ref 13,14 and 32, this study forms a very interesting set of new results in the field of frustrated magnetism, that should lead to new experimental and theoretical developments. I thus support the publication of this new version.

We thank the reviewer for their supportive comments, which highlight the potential of our paper to stimulate new developments in field of frustrated magnetism.

Reviewer 2:

I have had the opportunity to review the revised manuscript by Paddison et al "Emergent Order in the Kagome Ising Magnet Dy₃Mg₂Sb₃O₁₄." I am satisfied by the response to the referee reports and I agree that the manuscript should be transferred to Nature Communications. One of the referees point out that there is "no new physics" in this work due to previous work that just came out by Dun et al, but I disagree with this statement. It is unusual to observe emergent charge ordering in a kagome material, and the neutron scattering results presented here are important for showing this explicitly. I recommend publication in Nature Communications.

We are very pleased that the reviewer is satisfied by our responses to their previous comments, and recommends publication in Nature Communications.